# Theoretical Study of Microwires with an Inhomogeneous Magnetic Structure Using Magnetoimpedance Tomography

**DOI:** 10.3390/s24113669

**Published:** 2024-06-05

**Authors:** Nikita A. Buznikov, Galina V. Kurlyandskaya

**Affiliations:** 1Institute for Theoretical and Applied Electromagnetics, Russian Academy of Sciences, Moscow 125412, Russia; 2Institute of Natural Sciences and Mathematics, Ural Federal University, Ekaterinburg 620002, Russia; galinakurlyandskaya@urfu.ru

**Keywords:** magnetoimpedance, amorphous microwires, modeling, magnetic structure, magnetic anisotropy, permeability

## Abstract

The recently proposed magnetoimpedance tomography method is based on the analysis of the frequency dependences of the impedance measured at different external magnetic fields. The method allows one to analyze the distribution of magnetic properties over the cross-section of the ferromagnetic conductor. Here, we describe the example of theoretical study of the magnetoimpedance effect in an amorphous microwire with inhomogeneous magnetic structure. In the framework of the proposed model, it is assumed that the microwire cross-section consists of several regions with different features of the effective anisotropy. The distribution of the electromagnetic fields and the microwire impedance are found by an analytical solution of Maxwell equations in the particular regions. The field and frequency dependences of the microwire impedance are analyzed taking into account the frequency dependence of the permeability values in the considered regions. Although the calculations are given for the case of amorphous microwires, the obtained results can be useful for the development of the magnetoimpedance tomography method adaptation for different types of ferromagnetic conductors.

## 1. Introduction

The magnetoimpedance (MI) effect consists in significant change in the complex impedance of a ferromagnetic conductor under application of an external magnetic field [1,2,3]. MI is promising for the development of magnetic sensors and magnetometers with very high sensitivity with respect to applied magnetic fields [4,5,6,7,8], including devices for the detection of biomagnetic signals and magnetic biomarkers [9,10,11,12]. The nature of MI is attributed to the dependence of the skin penetration depth of a magnetic conductor on the external magnetic field. A significant MI effect can be observed in different soft magnetic amorphous and nanocrystalline as well as in complex composites [13,14,15,16].

In addition, the microwave properties of glass-coated microwires have attracted particular interest. In some cases, the developed metamaterials have shown extraordinary characteristics that are interesting for high-frequency applications in various devices [17]. Figure 1 shows selected examples of different types of magnetic wires demonstrating promise for MI applications. The varieties of their magnetic, magnetoelastic and microwave properties open the possibility to create modern magnetic sensors of multifunctional type adapted to different temperature ranges and environmental conditions. For example, water-quenched amorphous wires of selected composition and glass-coated microwires (Figure 1a–d) can be very good candidates for applications in biological and harsh chemical conditions. Figure 1 shows images obtained by scanning electron microscopy (SEM) representing different features of the samples. The details of the SEM technique [18] or preparation of the rapidly quenched wires [2,19], glass-coated microwires [20] and electroplated wires [21] can be found elsewhere. However, it is important to emphasize that the wire samples used for imaging are not exactly the same as those used for the magnetic and MI measurements or in a particular sensor device. Magnetic wires of all types have a certain disadvantage: it is necessary to use a special procedure in order to cut them without deformation of the edge part and install them into electronic circuit. Figure 1a,b show purposely deformed rapidly quenched wire in a way that one can appreciate the evidence of the amorphous structure. Figure 1c,d show a glass-coated microwire with broken glass shell, making it possible both to estimate the geometrical parameter of each part of the composite sample and evaluate the quality of the metal surface. Figure 1e is the same, but in the case of an electroplated wire.

There are two different types of magnetic biosensors: one for magnetic label detection, and an electrochemical type of detector that works on the label-free principle [22,23]. For magnetic label detection, the MI-sensitive element should be stable in biological and harsh chemical conditions. This can be achieved not only by the selection of appropriate composition, but also by coating/protecting the surface of the sensitive element with an additional metallic, polymer or carbon layer to make it biocompatible [24,25,26,27,28]. Even more, magnetic biosensors are usually designed with an additional layer either for functionalization of the sensitive element or for surface protection [24,25]. According to traditional classification of magnetic biosensors, the first type is for the the measurement of parameters of electrical and magnetic properties of living systems related to their functionality; the biocompatibility does not require the direct contact of the biosystem with the surface of the sensitive element [29,30,31]. The second type of magnetic biosensors (for testing the bioanalytes) is described above. In addition, electroplated wires having a nanostructured magnetic layer might be suitable for aerospace applications or other rugged radiation conditions [16].

One of the most-studied MI materials is glass-coated amorphous microwires produced by the Taylor–Ulitovsky technique [32,33,34,35,36,37,38]. In these microwires, magnetocrystalline anisotropy is negligible and the magnetoelastic anisotropy is governed by the magnetostriction and spatial distribution of internal stresses arising in amorphous metal during the glass-coated microwire production [32,34,35,36]. Uneven temperature distribution during the microwire fabrication and the difference in the thermal expansion coefficients of the glass covering and metal core parts leads to a non-uniform distribution of internal stresses over the microwire cross-section [32,35]. The magnetoelastic anisotropy in amorphous microwires can be modified by different post-annealing treatments [36,38,39], which results in a relaxation of the internal stresses.

Several models based on the theory of viscoelasticity have been proposed in order to describe the internal stress distribution in glass-coated amorphous microwires (see, for example, [35,40,41,42,43]). Usually, the magnetic structure of the amorphous part of glass-coated microwire is described in terms of the core–shell model [32]. It is assumed that the internal core has an axial magnetic anisotropy, whereas the external shell has a circular or radial easy axis of magnetization depending on the sign of the magnetostriction coefficient. The core–shell representation of the magnetic domain structure of glass-coated microwires allows one to explain peculiarities of their magnetic behavior. However, amorphous microwires may have a more complex domain structure. In particular, the magneto-optical Kerr effect studies for glass-coated microwires with a nearly-zero magnetostriction coefficient have shown that a helical anisotropy does exist in a surface region [44,45]. In addition, it has been demonstrated that the domain wall between the core and shell regions may significantly affect the magnetic behavior of microwires with a nearly-zero magnetostriction coefficient [46].

The use of a glass-coated amorphous microwire as an MI-sensitive element requires sufficient understanding of its magnetic anisotropy and magnetic structure, since the MI response depends on the microwire geometry, composition and post-preparation annealing/cooling or other type of additional treatment. In the first approximation, the impedance of a magnetic microwire can be described as follows [47]:(1)Z/Rdc=(ka/2)J0(ka)/J1(ka).
Here, *R*_dc_ = *l*/*πσa*^2^ is the microwire resistance in the direct current mode; *l* and *a* are the length and radius of the amorphous part of the microwire; *σ* is the conductivity; *J*_0_ and *J*_1_ are the Bessel functions of the first kind of zero and first orders; *k* = (1 + *i*)/*δ*; *i* is the imaginary unit; *δ* = *c*/(2*πσωμ*)^1/2^; *c* is the speed of light in vacuum; *ω* is the angular frequency and *μ* is the circumferential permeability.

Equation (1) was obtained assuming that the circumferential permeability does not change over the microwire cross-section. Moreover, the tensor form of the permeability is neglected in Equation (1). The models for MI responses taking into account the tensor form of the permeability were developed for the geometry of a cylindrical conductor with circular or helical anisotropy [48,49,50]. It was shown that these models are applicable in cases of asymmetric MI and off-diagonal MI effect.

The influence of the axially magnetized core on the MI in the rapidly quenched amorphous wires and glass-coated microwires was investigated in the framework of different models [51,52,53,54,55]. It was demonstrated that in some cases, the core–shell models allow one to explain specific features of the frequency dependences of the MI response.

It should be noted that there are no methods for the direct study of the magnetic structure in the cross-section of glass-coated amorphous microwires, so it is usually interpreted on the basis of magnetostatic measurements. Recently, a new method to evaluate magnetic structure in amorphous materials of different types exhibiting a MI effect was developed [56,57,58,59,60,61,62]. The method is based on measured data on the frequency dependence of the MI, which allows one to analyze the distribution of the permeability over the conductor cross-section. This method is referred to as MI tomography.

In Ref. [56], the radial distribution of the permeability in glass-coated amorphous microwires was restored by using Equation (1). The permeability was reconstructed by averaging over a surface layer of the microwire to which part the major intensity of the excitation current corresponded. However, this approach has some limitations when the permeability has strongly non-uniform distribution over the microwire cross-section [56].

Another approach is based on a comparison of the experimental frequency dependences of the MI with numerical simulation [57]. The conductor cross-section is divided into several regions with constant electromagnetic properties. The Maxwell equations were solved numerically by the finite element method by using Comsol Multiphysics commercial software (https://www.comsol.com/products). The method allowed description of the experimental data on the MI frequency dependences in the composite CuBe/FeCoNi wires [57,58,59], amorphous rapidly quenched wires [57,60,62] and amorphous ribbons [61].

Note that there are some assumptions in the method developed in Ref. [57]. First, the field dependence of the permeability in each region was not specified explicitly. Second, it was assumed that the permeability is independent of the frequency. In addition, the imaginary part of the permeability was neglected. Although, in many cases, these approximations are well applicable, it is of interest to develop a model that does not have these restrictions.

In this work, we propose a theoretical model for the description of the MI effect in amorphous microwires with inhomogeneous magnetic structure. The microwire cross-section is divided into several regions, where the magnetic properties are assumed to be constant. The approach is based on the calculation of electromagnetic fields in the regions of the microwire by using an analytical solution of Maxwell equations. In the framework of the model, the frequency dependence and complex nature of the permeability are taken into account, and the field dependence of the permeability is expressed in an explicit form.

## 2. Model

Let us consider amorphous microwire exciting by the alternating current *I*(*t*) = *I*_0_exp(–*iωt*). The external magnetic field *H_e_* is directed along the microwire axis. Following the approach proposed previously [57], we divide the amorphous nucleus of the microwire into *n* coaxial regions. It is assumed that each region has uniaxial magnetic anisotropy, and the anisotropy axis makes an angle *ψ_j_* with the azimuthal direction (hereinafter, the index *j* = 1, … *n* denotes the region number in the model).

To find the magnetization distribution, we neglect the exchange and magnetostatic coupling between the selected regions. This approximation significantly simplifies the calculations. However, it should be noted that the coupling between the regions can be taken into account by introducing effective interaction fields similar to the core–shell models proposed previously [53,54].

The static equilibrium magnetization distribution within the regions can be found by the minimization of the free energy. The free energy density *U* for each region can be presented as a sum of the magnetic anisotropy term and Zeeman energy [13]:(2)U=(MHj/2)sin2(θj−ψj)−MHesinθj.
Here, *M* is the saturation magnetization, *H_j_* is the anisotropy field in the region *j* and *θ_j_* is the equilibrium magnetization angle with respect to the azimuthal direction.

The minimization of the free energy density results in the following equations for the equilibrium magnetization angles *θ_j_*:(3)Hjsin(θj−ψj)cos(θj−ψj)=Hecosθj.

In the following analysis, we neglect the contribution of the domain-wall motion to the permeability and assume that the values of the permeability in the regions are governed by the magnetization rotation. This approximation is valid at not-too-low frequencies, when the domain-wall motion is damped by eddy currents. In soft magnetic amorphous materials exhibiting the MI effect, the domain-wall motion is negligible in the frequency range from several hundred kHz to a few MHz [63,64]. It is assumed also that the permeability tensor has a quasi-diagonal form, and the MI response depends only on the values of the circumferential permeability in the regions. For simplicity, we neglect the contribution of the exchange-conductivity effect to the permeability in the regions. This contribution is relatively low at high frequencies, and it could be taken into account by means of the methods developed previously [49,65]. Under the above assumptions, the circumferential permeability *μ_j_* in the region *j* is expressed as [13,50,63]:(4)μj=1+γ4πM(γ4πM+ωj*−iκω)sin2θj(γ4πM+ωj*−iκω)(ωj**−iκω)−ω2.
Here, *γ* is the gyromagnetic constant, *κ* is the Gilbert damping parameter and the characteristic frequencies are given by [50]
(5)ωj*=γ[Hjcos2(θj−ψj)+Hesinθj] ,ωj**=γ[Hjcos{2(θj−ψj)}+Hesinθj] .

Taking into account the cylindrical symmetry, the distribution of electromagnetic fields within the microwire is described by Maxwell equations [47]:(6)−∂ej∂ρ=iωμjchj ,1ρ∂∂ρ(ρhj)=4πσcej .
Here, *ρ* is the radial coordinate; *e_j_* and *h_j_* are the amplitudes of the longitudinal electric field and the circular magnetic field, respectively.

The solution of Equation (6) for each region is expressed as follows:(7)ej=(ckj/4πσ)[AjJ0(kjρ)+BjY0(kjρ)] ,hj=AjJ1(kjρ)+BjY1(kjρ) .
Here, *k_j_* = (1 + *i*)/*δ_j_*; *δ_j_* = *c*/(2*πσωμ_j_*)^1/2^; *Y*_0_ and *Y*_1_ are the Bessel functions of the second kind of zero and first orders; *A_j_* and *B_j_* are the constants. Note that due to the symmetry, the constant *B*_1_ is equal to zero in the central region, *j* = 1.

The constants *A_j_* and *B_j_* can be found from the continuity conditions for the amplitudes of the electric and magnetic fields at the interfaces between different regions, *ρ* = *r_j_* (*j* < *n* − 1):(8)ej(rj)=ej+1(rj) ,hj(rj)=hj+1(rj) .

Using Equation (7), these conditions can be rewritten as
(9)AjJ0(kjrj)+BjY0(kjrj)=(μj+1/μj)1/2[Aj+1J0(kj+1rj)+Bj+1Y0(kj+1rj)] ,AjJ1(kjrj)+BjY1(kjrj)=Aj+1J1(kj+1rj)+Bj+1Y1(kj+1rj) .

The boundary condition at the surface of amorphous part of the microwire is obtained from the current excitation condition. For the surface region, *j* = *n*, we have
(10)AnJ1(kna)+BnY1(kna)=2 I0/ca.

The field distribution within the microwire is completely determined by Equations (7), (9) and (10). The microwire impedance *Z* is expressed as [48,49,50]
(11)Z=len(a)I0=2lca×en(a)hn(a),
where *e_n_*(*a*) and *h_n_*(*a*) are the amplitudes of the electrical and magnetic field at the surface of the amorphous part of the microwire.

Equation (11) can be rewritten as follows:(12)Z/Rdc=(kna/2)×AnJ0(kna)+BnY0(kna)AnJ1(kna)+BnY1(kna).

Note that for a uniform distribution of the anisotropy over the microwire cross-section, *n* = 1, we have *B_n_* = 0 from the symmetry conditions. Then, Equation (9) transfers to a standard expression for the microwire impedance (see Equation (1)).

Thus, the procedure for the calculation of the MI response in the microwire with inhomogeneous magnetic structure can be summarized as follows. The cross-section of the metallic part of the microwire is divided into *n* regions. In principle, there are no restrictions on the selection of the number of regions. For each region, the anisotropy field *H_j_* and the anisotropy axis *ψ_j_* can be specified. The equilibrium magnetization angles *θ_j_* and the circumferential permeability *μ_j_* in the region *j* are found from Equations (3) and (4), respectively. After that, the field distribution within the microwire is calculated using Equations (7), (9) and (10), and the impedance *Z* is obtained by means of Equation (12).

## 3. Results and Discussion

The proposed model allows one to analyze the MI effect in amorphous microwires with inhomogeneous magnetic structure. A comparison of modeling results with measured data on the MI in microwires is out of the scope of this paper, and below we present some results of the MI calculations to illustrate the applicability of the model. For calculations, we use the following parameters of amorphous microwire: the diameter of metallic nucleus 2*a* = 20 μm, the saturation magnetization *M* = 600 G, the conductivity *σ* = 10^16^ s^−1^ and the Gilbert damping parameter *κ* = 0.1.

We divide the cross-section of the amorphous part of the microwire into five regions, *n* = 5. It is assumed that in the central region, the easy magnetization axis is along the microwire axis. In the surface region, the anisotropy is close to the circular one. In the intermediate regions, the anisotropy axes change gradually from the axial to transverse directions. The radii of the regions *r_j_*, the anisotropy fields *H_j_* and the anisotropy axis angles *ψ_j_* are presented in Table 1.

The dependences of the values of the static permeability in the regions on the external field are shown in Figure 2. The values of *μ_j_* are calculated by means of Equations (3)–(5) at *ω =* 0 using the parameters presented in Table 1.

The permeability in two central regions, *j* = 1 and *j* = 2, has a maximum at zero external field and decreases with a growth of *H_e_*. In other regions, the permeability has non-monotonic behavior and achieves a peak at some non-zero external field. The difference in the field dependences of the permeability values significantly affect the MI response of the microwire. Note that at high external fields, the permeability values for different regions approach each other (see Figure 2). This is due to the fact that the magnetization directions in each region are close to the direction of the external field. The magnetization distribution over the microwire cross-section becomes almost uniform, and the permeability values depend weakly on the anisotropy field.

Let us return to Equation (4) and results shown in Figure 2. The developed model for calculation of the frequency dependence and complex nature of the magnetic permeability were taken into account. However, Figure 2 presents values of the static permeability to illustrate the changes in the permeability in the regions with external field just for the simplicity of visual representation.

Further, we compare the results of the MI modeling by using the five-region approach, *n* = 5, with two other models for distribution of the magnetic anisotropy within the microwire. In the first model, it is assumed that the anisotropy distribution is uniform over the microwire cross-section, *n* = 1. Then, the MI response can be found by using Equation (1). It is assumed that in this case, the anisotropy field and the anisotropy axis angle coincide with those for the surface region in the five-region model, that is, *H*_1_ = 20 Oe and *ψ*_1_ = 0.05*π*.

In the second model, we assume that the magnetic structure of the microwire is described in the framework of the core–shell model, *n* = 2. Note that for *n* = 2, the microwire impedance *Z* is expressed in the following form [51]:(13)Z/Rdc=(k2a/2)×J0(k2a)+PY0(k2a)J1(k2a)+PY1(k2a) ,P=(μ2/μ1)1/2J1(k1r1)J0(k2r1)−J0(k1r1)J1(k2r1)J0(k1r1)Y1(k2r1)−(μ2/μ1)1/2J1(k1r1)Y0(k2r1) .
Here, *r*_1_ is the radius of the inner core, and the subscripts 1 and 2 correspond to the core and shell regions. Further, we take the following values of parameters for calculations in the core–shell model: 2*r*_1_ = 10 μm, *H*_1_ = 5 Oe, *H*_2_ = 20 Oe, *ψ*_1_ = 0.5*π* and *ψ*_2_ = 0.05*π*.

The studied magnetic structure with axial anisotropy in the inner region of the microwire and close to the circular anisotropy in the surface region is typical for Co-rich glass-coated amorphous microwires with nearly-zero or negative magnetostriction [13,32]. These microwires exhibit the highest values of the MI effect. A similar structure exists in stress-annealed Fe-rich glass-coated amorphous microwires with positive magnetostriction, since the circular anisotropy appears in the surface region after the annealing [36,38,55,66,67]. Note also that the proposed model is applicable for microwires with a regular domain structure in the surface layer consisting of ring domains (so-called bamboo domain structure), since Equation (4) is valid for description of the effective circumferential permeability after averaging over the surface domain structure.

Figure 3 shows the field dependences of the MI ratio Δ*Z*/*Z* calculated at several frequencies *f* = *ω*/2*π* for different models. Note that the dependences of Δ*Z*/*Z* on *H_e_* is presented only for positive values of the external magnetic field, since in the framework of the model, the field dependences of the MI ratio are symmetric with respect to the sign of the external field. The MI ratio is defined as follows:(14)ΔZ/Z=[Z(He)−Z(H0)]/Z(H0),
where *H*_0_ = 150 Oe is the external field sufficient for magnetic saturation of the microwire.

It follows from Figure 3 that at relatively low frequencies of the excitation current, the discrepancy between the calculated results for the field dependence of Δ*Z*/*Z* ratio in the models with different number of regions *n* is quite high. The model with uniform magnetic anisotropy distribution, *n* = 1, predicts that the field dependence of the MI ratio exhibits two-peak behavior. For the core–shell model, *n* = 2, the MI ratio has an additional maximum at zero field (see Figure 3a,b). This fact is related to the contribution of the core region with high permeability at low fields. The five-region model, *n* = 5, provides the results intermediate between *n* = 1 and *n* = 2 at low fields.

With an increase in the frequency, all the models give almost identical results, with the exception of the field region close to zero magnetic field (Figure 3c,d). This is due to the fact that at high frequencies, the thickness of the skin layer decreases, and the main contribution to the MI comes from the surface region, which has the same magnetic properties for all the models under consideration.

The MI tomography method is based on a comparison of calculations of the frequency dependence of the MI with measurement results. The calculation results are very sensitive to the distribution of the permeability over the microwire cross-section. Figure 4 presents the frequency dependences of the impedance *Z* calculated at several external fields for different models. At zero external field, the difference between the three models is of significance at all frequencies (Figure 4a). The discrepancy between the calculated impedance values for different numbers of regions decreases with an increase in the external field. At *H_e_* > 10 Oe, all the studied models provide very similar impedance values within the whole frequency range, since the permeability values are approximately the same for all regions at high external fields. The obtained results are in qualitative agreement with numerical calculations using the finite element method [57]. Thus, to study the anisotropy distribution within the microwire by the MI tomography method, the frequency dependence of the impedance at low external fields should be analyzed.

In conclusion of this section, let us discuss the frequency dependence the MI and the effect of the imaginary part of magnetic permeability on the MI response. In the framework of the model proposed, the absence of the frequency dependence of the circumferential permeability can be obtained by setting *ω* = 0 in Equation (4). The imaginary part of the permeability is neglected if we assume in Equation (4) that *ω* ≠ 0 and *κ* = 0.

An analysis shows that at all frequencies, the highest difference between the proposed model and simplified approaches is observed near the maximum of the field dependence of the impedance. In this regard, we analyze further the frequency dependence of the maximum MI ratio (Δ*Z*/*Z*)_max_ calculated for the cases when the frequency dependence of the permeability is neglected, *ω* = 0, and when the imaginary part of the permeability equals zero, *κ* = 0. The calculated frequency dependences of (Δ*Z*/*Z*)_max_ are shown in Figure 5 for the five-region model, *n* = 5.

Figure 5 shows that for the studied microwire at frequencies below 70 MHz, the difference in the calculated values of (Δ*Z*/*Z*)_max_ is negligible. The significant difference between different models appears at the frequencies higher than 100 MHz, and it increases with the frequency. Note that the frequency dependence of maximum MI ratio exhibits different behavior in the proposed model and simplified approaches.

In the model proposed, the maximum MI ratio has a peak and decreases at high frequencies (see curve 1 in Figure 5). If the frequency dependence of the permeability is not taken into account, then the maximum MI ratio is almost constant within a wide frequency range (see curve 2 in Figure 5). When we take into account the frequency dependence of the permeability and neglect the imaginary part of the permeability, then (Δ*Z*/*Z*)_max_ increases with the frequency (see curve 3 in Figure 5). Thus, the analysis demonstrates that the simplified approach developed previously [57,58,59,60,61,62] is valid within a wide frequency range. However, at high frequencies, the frequency dependence and the complex nature of the circumferential permeability should be taken into account in order to describe the MI response of amorphous microwires correctly.

Despite the main focus of the present study being on glass-coated magnetic microwires, the obtained results are not limited by this particular kind of MI materials, and the developed model can be useful for MI tomography as an additional instrument complementing existing methodology. For example, a recently proposed type of biphase glass-coated microwires, i.e., composites based on magnetic microwire having an additional magnetic coating deposited by the sputtering technique would be of special interest to researchers and engineers. Such a phenomenon as low field microwave absorption or low field MI [16,37,68,69,70] to some extent could be rethought and may be even better understood.

In the present work we made a focus on the study of glass-coated microwires. Future research directions can be connected to the development of this approach for the other cylindrical ferromagnetic conductors and even more complex composites.

## 4. Conclusions

The MI tomography method consists in a comparison of the calculated and measured frequency dependences of the MI at different external fields. The method can be used to restore the distribution of the magnetic permeability and magnetic anisotropy over the conductor cross-section.

In this work, we designed, proposed and tested a theoretical model for the description of the MI effect in amorphous microwires with inhomogeneous magnetic structure over their cross-section. This approach allows one to analyze the distribution of magnetic properties over the cross-section of the conductor, including the cases of composite conducting materials. The microwire cross-section is divided in several regions, in which the magnetic properties are assumed to be constant. The approach is based on the calculation of electromagnetic fields in the regions of the microwire by using an analytical solution of the Maxwell equations. The frequency dependence and complex nature of the permeability are taken into account, and the field dependence of the permeability is expressed in an explicit form in the framework of the model. The results obtained can be useful for the application and further development of the MI tomography method.

## Figures and Tables

**Figure 1 sensors-24-03669-f001:**
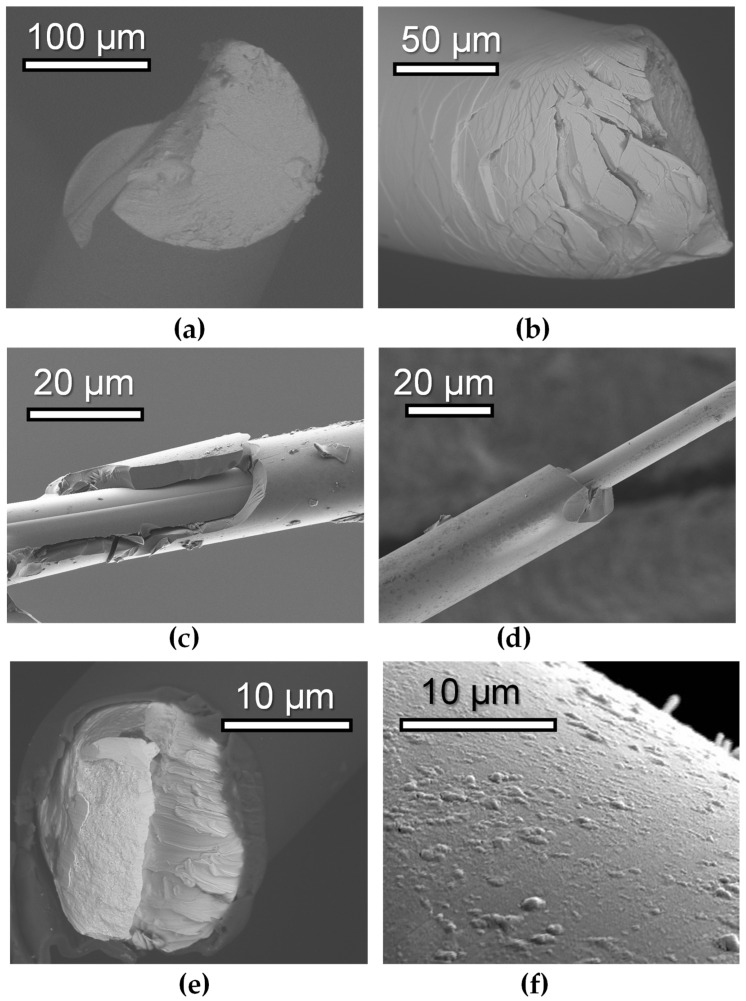
SEM images showing the general views of different types of magnetic wires. Cold-drawn in the rotating water amorphous wires with following compositions: Fe_75_Si_10_B_15_ (**a**) and (Co_94_Fe_6_)_72_._5_Si_12_._5_B_15_ (**b**). Glass-coated amorphous microwires obtained by Taylor–Ulitovsky technique: (Co_94_Fe_6_)_72_._5_Si_12_._5_B_15_ (**c**) and (Co_50_Fe_50_)_72_._5_Si_12_._5_B_15_ (**d**). Composite CuBe/CoFeNi electroplated wires: cross-section showing both the central conductive base wire and magnetic coating (**e**), general features of the surface properties of CoFeNi layer showing different types of the surface defects appearing daring the electroplating process (**f**).

**Figure 2 sensors-24-03669-f002:**
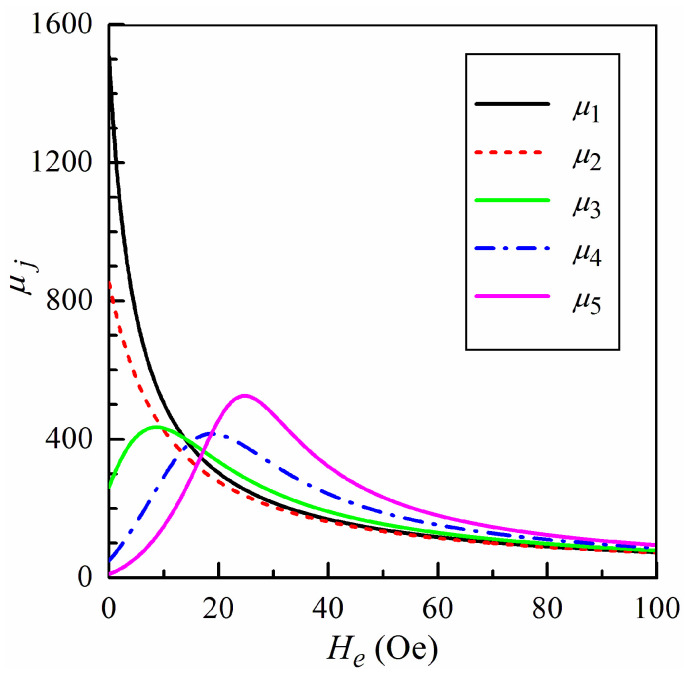
The values of the static permeability in the five-region model, *n* = 5, versus the external field *H_e_*.

**Figure 3 sensors-24-03669-f003:**
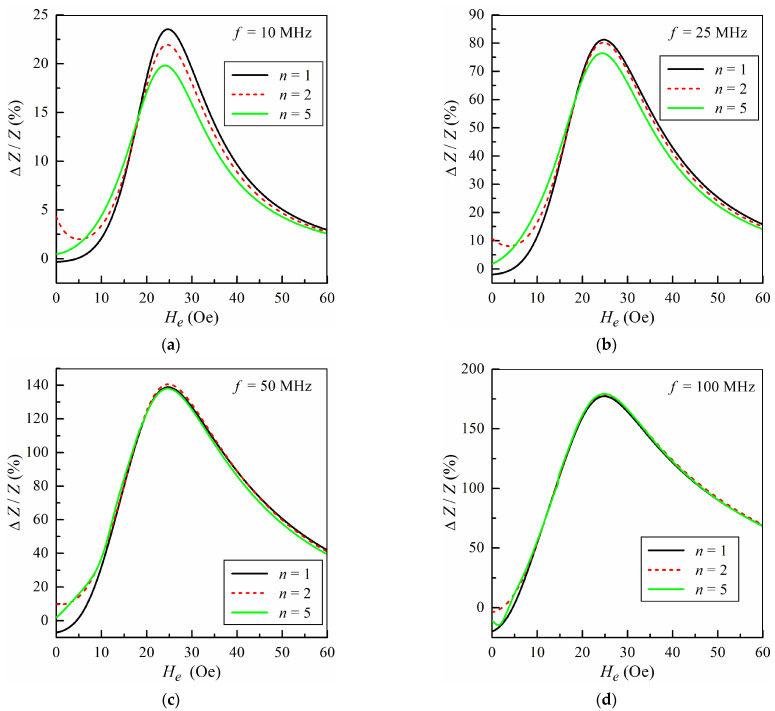
The field dependences of the MI ratio Δ*Z*/*Z* calculated for different numbers of regions *n* at several frequencies: *f* = 10 MHz (**a**); *f* = 25 MHz (**b**); *f* = 50 MHz (**c**); *f* = 100 MHz (**d**).

**Figure 4 sensors-24-03669-f004:**
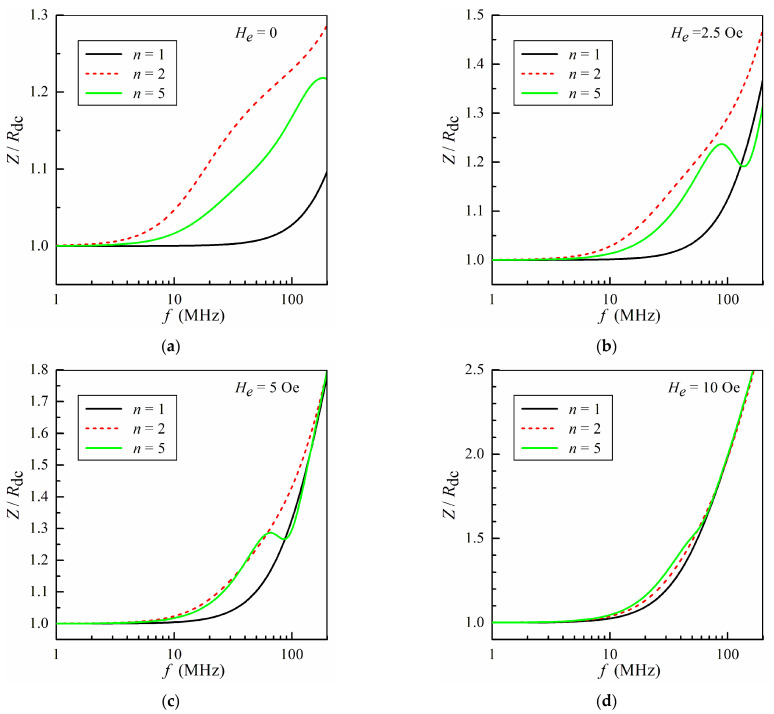
The frequency dependence of the impedance *Z* calculated for different numbers of regions *n* at several external magnetic fields: *H_e_* = 0 (**a**); *H_e_* = 2.5 Oe (**b**); *H_e_* = 5 Oe (**c**); *H_e_* = 10 Oe (**d**).

**Figure 5 sensors-24-03669-f005:**
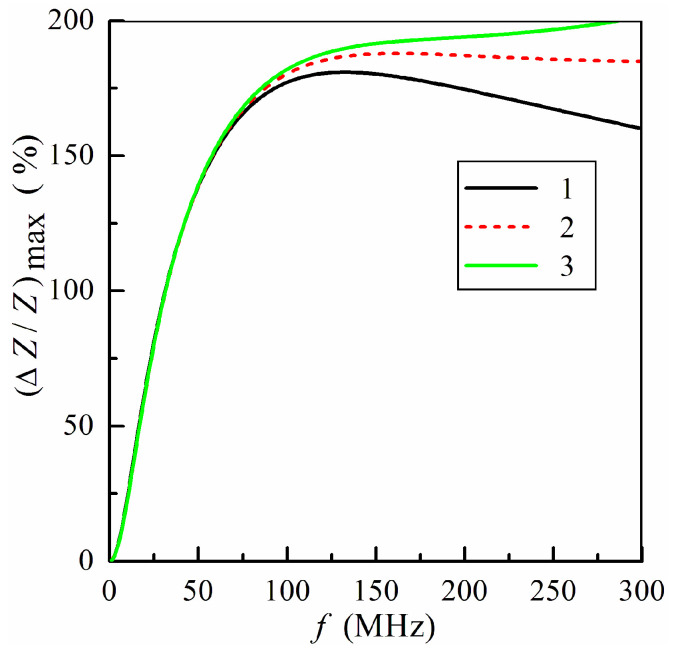
Frequency dependence of maximum MI ratio (Δ*Z*/*Z*)_max_ for the number of regions *n* = 5: curve 1, the proposed model; curve 2, model with the permeability independent of the frequency; curve 3, model with the imaginary part of the permeability equal to zero.

**Table 1 sensors-24-03669-t001:** Parameters of the five-region model, *n* = 5.

*j*	Region Radius *r_j_*, μm	Anisotropy Field *H_j_*, Oe	Anisotropy Axis Angle *ψ_j_*, rad
1	2	5	0.5*π*
2	4	8	0.4*π*
3	5	10	0.2*π*
4	7	15	0.1*π*
5	10	20	0.05*π*

## Data Availability

Data available from the corresponding author upon reasonable request.

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
