# Peer review of "Theoretical Study of Microwires with an Inhomogeneous Magnetic Structure Using Magnetoimpedance Tomography"

_sensors, 2024, doi:10.3390/s24113669_

Round 1
Reviewer 1 Report
Comments and Suggestions for Authors
The work “Magnetoimpedance Tomography of Amorphous Microwires with an Inhomogeneous Magnetic Structure as a Novel Approach to the Properties Evaluation” is devoted to the description of an example of theoretical study of the magnetoimpedance effect in an amorphous microwire with inhomogeneous magnetic structure. The results of the work are of significant interest and can find application in the further development of magnetic impedance tomography of the magnetic domain structure and properties of microwires with cylindrical symmetry. The work can be published in the journal Sensors after a major revision and elimination of a number of deficiencies after responding to the following comments:
1. In the Introduction section, the authors write about the studied microwires (lines 43-45): “For example, the in-water-quenched amorphous wires of selected composition and glass-coated microwires (Figure 1 (a)–(d)) can be very good candidates for their applications in biological and harsh chemical conditions.” It is not very clear exactly how wires with composition of Fe75Si10B15 can be used in biological applications. The composition of such wires is not biologically compatible. In addition, these wires do not have a glass shell, which could be biocompatible (for example, it can be made from Pyrex). Authors should clarify this point in the manuscript.
2. In the Introduction section, the authors also write (lines 100-102): “It should be noted that there are no methods of direct study of the magnetic structure in glass-coated amorphous microwires, and usually it is interpreted on the basis of magnetostatic measurements". This is not entirely correct, since the surface domain structure of such objects can be studied using the magneto-optical Kerr effect. By the way, the authors also write about this above. Obviously, the authors meant a direct study of the magnetic structure in the cross-section of a microwire along its entire length. This point needs to be clarified.
3. In the Model section, the authors write (lines 154-155): “This approximation is valid at not too low frequencies, when the domain-walls motion is damped by eddy currents.” Despite the fact that this is a known phenomenon, it is necessary to at least roughly indicate the interval or lower limit of such frequencies.
4. The model described in the work is applicable for microwires with cylindrical symmetry (Figures 1c, 1d). However, it cannot be applied to microwires with distorted cylindrical symmetry (Figures 1a, 1e). Thus, the constant B1 cannot be used equal to zero in expression (7), just as other simplifications become inapplicable. This must be explicitly stated in the text of the work.
5. Also, from the description of the model, one gets the impression that this model is applicable only to microwires with nearly-zero or (with some assumptions) negative magnetostriction. At the same time, it is difficult to assume that this model is applicable for microwires with a ring domain structure of the surface layer and positive magnetostriction. This point should be noted in the work.
6. It is not completely clear how the model presented in this work is combined with the ring domain structure of the surface layer, consisting of ring domains with a changing orientation of the magnetic moment (circular or radial). As is known, such domains can have a variable width along the wire, that is, the structure can be non-periodic. Some discussion on this topic could brighten the work.
7. The authors chose the Gilbert damping parameter equal to 0.1 (line 200). Please clarify for what reasons or on the basis of what references this particular parameter value was chosen?
8. In Figure 3, the authors present the field dependences of the MI ratio ΔZ/Z calculated for different numbers of regions n at several frequencies. Typically, as the frequency of the flowing current in microwires increases, a gradual shift of the impedance peak to the right (in the direction of increasing field) is observed. In Figure 3 it is difficult to talk about the existence of such a shift. What limitations of the considered models may this be associated with? An explanation of this phenomenon could expand the results of the work.
9. From the description of the proposed model, it is clear that the distribution of the electric and magnetic fields over the volume was included in the model itself. The question arises: was the change in the depth of the skin layer considered when analyzing the modeling results? This is quite simple to do using the models described in the Introduction and given the estimated values of circular magnetic permeability.
Comments on the Quality of English LanguageThere are some poor formulations that could be improved.
Reviewer 2 Report
Comments and Suggestions for Authors
The current manuscript describes a theoretical study on applying magnetoimpedance tomography (MIT) to analyze the magnetic properties of amorphous microwires with an inhomogeneous structure. After creafully reading I find the work is clearly states the focus on the recently proposed MIT method and Highlights the capability of MIT to analyze magnetic property distribution within a conductor. In addtiono, mentions the specific example of an amorphous microwire with a non-uniform magnetic structure, describes the model used, involving different regions with varying effective anisotropy, touches upon the analytical solution for electromagnetic fields and impedance calculations. Finally emphasizes the frequency dependence of permeability and its consideration in the análisis and broadens the applicability of the results beyond amorphous microwires. But there are some point needs to be calrify before accepting the current manuscript in Sensors:
1- Could the authors describe in details about the specific features of the chosen anisotropy distribution.
2- The authors didnot explicitly mention the type of analysis performed on the field and frequency dependence of impedance.
3- Could the authors benefit from a sentence mentioning potential limitations of the model or future research directions.
Minor issues:
- Title: I agree, the title is too long. Consider the suggested options:
- "Analyzing Inhomogeneous Magnetic Structures in Microwires using Magnetoimpedance Tomography (MIT)" or "Unveiling Complexity: Magnetoimpedance Tomography for Microwires with Non-Uniform Magnetism"
- Introduction:
- The introduction needs significant improvement. It should:
- Clearly state the problem being addressed (importance of analyzing inhomogeneous magnetic structures).
- Briefly explain the limitations of existing methods (if applicable).
- Highlight the novelty and advantages of using Magnetoimpedance Tomography (MIT) for this purpose.
- Provide a brief overview of the research methods and applications.
- Figure 1:
- Clarify the source of the microwire systems in Figure 1.
- If fabricated by the authors, add a brief description of the experimental methods.
- If cited from other work, add references.
- Equations:
- Include proper citations for all equations (equations 1-14) used in the report.
- Conclusion:
- Expand the conclusion to include:
- The potential applications of the research findings.
- Future research directions based on the developed model.
Reviewer 3 Report
Comments and Suggestions for Authors
The article presents new results regarding amorphous microwires. The article cannot be published in its present form. Major revision is needed.
1) The title does not reflect the essence of the research conducted. Although the authors have already used the term tomography in articles related to microwires, readers may be misled into thinking that this is an experimental study. But that's not true. Here we see the application of a relatively simple model of layer-by-layer magnetic impedance modeling. This is exactly what the title should reflect.
“ Novel Approach” – It is not novel. The authors have already reported this method: Sensors 2022, 22, 9512.
“Properties Evaluation” – which properties? It is too general.
2) Figure 1 is not related to the studies presented in the article. The details of the microwire structure are not discussed at all. The figure 1 could be deleted without affecting the presented results.
“Figure 1 shows selected examples of different types of magnetic wires demonstrating promising for applications sensitivities of the MI effect.”
- It's difficult to understand what authors are talking about. “promising for applications sensitivities”?
3) “It should be noted that there are no methods of direct study of the magnetic structure in glass-coated amorphous microwires, and usually it is interpreted on the base of magnetostatic measurements.”
- This seems like an obvious exaggeration. The magnetic structure of microwires is interpreted on the basis of a whole range of experimental techniques, including high-frequency FMR, GMI, etc.
4) Introduction: “the frequency dependence and complex nature of the magnetic permeability are taken into account, and the field dependence of the permeability is expressed in an explicit form.”
- In fact, the authors do not present in the article “frequency dependence and complex nature of the magnetic permeability”. Presented only “values of the static permeability” (Fig. 2).
5) “For calculations, we use the following parameters of amorphous microwire: the diameter of metallic nucleus 2a = 20 μm, the saturation magnetization M= 600 G, the conductivity … and the Gilbert damping parameter k = 0.1.”
It is not clear why these particular parameters were chosen, and how changing them will affect the final version.
6) “We divide the cross-section of the amorphous part of the microwire into five regions, n = 5.”
Why not more, 10 for example?
7) Why exactly those parameters were chosen that are presented in Table 1. If the authors present this table in the form of the image of a cross-section of a microwire, the presentation will be more clear (change in anisotropy angle from 0.5 pi to 0.05 pi, and so on).
8) “The values of μj are calculated by means of Equations (3)–(5) at ω = 0”.
- A strange choice of ω = 0, considering that the article is studying high-frequency impedance.
9) Although the authors write that
“The MI tomography method is based on a comparison of calculations of the frequency dependence of the MI with measurement results”
nevertheless, they do not compare the results of calculations with experiments. They are compared with two other models, obviously simpler, which do not take into account the layered magnetic structure of the microwire. Why? Or did these other models take into account the same layering (see Fig. 5)?
10) In the text: “Figure 3 presents the frequency dependences of the impedance Z calculated at several external fields for different models.”
Figure 3 caption: The field dependences of the MI ratio ΔZ / Z calculated for different number of regions n at several frequencies.”
So, different models or different number of regions?
11) Why do the authors, claiming 5 regions, show results for only three (Figs. 3 and 4)?
12) If the authors claim the concept of tomography, it is necessary to show the total result as the interference of contributions of all five claimed regions.
13) “The obtained results are in a qualitative agreement with numerical calculations by using the finite element method [38]”.
What does it mean in this case that there is a qualitative agreement with another method that did not take into account the layering proposed in this work? Why is it important? Does this mean that the layering is not important?
14) “This is due to the fact that at high frequencies, the thickness of the skin layer decreases, and the main contribution to the MI comes from the surface region, which has the same magnetic properties for all the models studied.”
Does this mean that the authors changed the parameters “region boundary” from the table 1 when the frequency increased?
15) “Further, we take the following values of parameters for calculations in 234 the core–shell model: 2r1 = 10 μm, H1 = 5 Oe, H2 = 20 Oe, Ψ 1 = 0.5π and Ψ2 = 0.05π.”
Why were these particular parameters chosen? How will the changing of these parameters affect the results?
16) “The calculated frequency dependences of (ΔZ/Z)max are shown in Figure 5 for the five-region model, n=5."
and
“Figure 5. Frequency dependence of maximum MI ratio (ΔZ/Z)max for the number of regions n=5: curve 1, the proposed model; curve 2, model with the permeability is independent of the frequency; curve 3, model with the imaginary part of the permeability is equal to zero.”
Unclear. There were three models, all with five regions? What is "the proposed model"?
17) “If the frequency dependence of the permeability is not taken into account, then the maximum MI ratio is almost constant within a wide frequency range (see curve 2 in Figure 5).”
It's not surprising if “frequency dependence of the permeability is not taken into account”.
18) Conclusions: “The frequency dependence and complex nature of the magnetic permeability are taken into account”.
I don't see the figure of “The frequency dependence of the magnetic permeability” in the manuscript.
19) Conclusions: “The results obtained can be useful for the MI tomography method” or are the results presented in this article actually tomography method?
Comments on the Quality of English LanguageIn some places the text is unclear. English needs to be improved.
Reviewer 4 Report
Comments and Suggestions for Authors
Overall, the paper is well-focused, clear, and easy to read. The model is thoroughly described, and the application procedure is clearly outlined. The use of a theoretical model that simplifies the solution of more complex structures beyond the core-shell model can be applied to complex wires or electro-deposited multi-layers on conducting wires.
Although the model is clear, my main concern is its application for the efficient analysis of the magnetoimpedance of wires. Even fitting the core-shell model to a dataset is challenging due to the high number of degrees of freedom. Using a model with five regions introduces numerous parameters, and many combinations may yield very similar results. Nevertheless, it is an interesting contribution to the field.
While it is stated in the article that this is not the aim of the work, I would have liked to see its application to an experimental dataset to better compare its advantages against the core-shell model.
Additionally, I have the following suggestions:
• An appendix (or suplementary material) detailing the algebra to derive equations (7) would be helpful. The selection of constants for both Bessel equations for hj and ej and their relations is not straightforward.
• In the set of equations (8), I would explicitly state that it is valid for j<n-1.
• In line 190, should it refer to equations (2) and (3) or equations (3) and (4)?
• I suggest including the following reference as an analytical solution of the core-shell model for impedance calculation, which can serve as a comparison with the results obtained in equation 13: D.-X. Chen et al., Journal of Magnetism and Magnetic Materials 202 (1999) 385-396.
As a final comment, I suggest writing the equations in the International System of Units (SI), as it has become the most common practice in recent years, although the CGS system is still frequently used in the field of magnetism.
Round 2
Reviewer 1 Report
Comments and Suggestions for Authors
The work “Theoretical Study of Microwires with an Inhomogeneous Magnetic Structure Using Magnetoimpedance Tomography” is devoted to the description of an example of theoretical study of the magnetoimpedance effect in an amorphous microwire with inhomogeneous magnetic structure. The results of the work are described in sufficient detail and do not raise objections. I am completely satisfied with the responses to the comments and the changes made to the work. The work can be published in the journal Sensors without further changes. I thank the authors for their interesting and thought-provoking work.
Reviewer 2 Report
Comments and Suggestions for Authors
The authors fix all my comments correctly and the current manuscript can be published in Sensors in present form.
Reviewer 3 Report
Comments and Suggestions for Authors
No comments. It could be published.